# OpenReview forum: "Fully Dynamic Coreset Spectral Clustering"
_ICLR.cc/2026/Conference — Submitted to ICLR 2026_

### Official Review · Reviewer_WXXY · 2025-10-22

**Soundness:** 3
**Presentation:** 2
**Contribution:** 3
**Rating:** 4
**Confidence:** 4

**Summary:**

This paper presents a new algorithm for spectral clustering in a dynamic graph. This builds upon the work of Jourdan et al, who recently showed how to maintain a coreset for a so-called "graph kernel" so that normalized cut problem on G can be approximated by running spectral clustering on the coreset. The idea of a graph kernel dates back to earlier work by Dhillon et al, who showed that the normalized cut objective and kernel k-means are equivalent, in the sense that normalized cut on a graph can be reformulated as kernel k-means for an appropriate choice of kernel.

The main contribution of this paper is to use the (static graph) results of Jourdan et al to develop an algorithm for dynamic graphs, but dynamically maintaining the coreset. At the heart of Jourdan et al's algorithm is a sampling tree that is used for sampling subsets of nodes. A naive dynamic algorithm would be to construct the sampling tree from scratch with each change to the graph, but this would be prohibitively expensive as it takes linear time to generate the sampling tree. The paper introduces a notion of "Just in time sampling trees" to address this limitation. The new approach have a favorable tradeoff in terms of update time and query time, relative to other dynamic approaches.

**Strengths:**

This paper has a lot of strengths:
* The problem of clustering in dynamic graph is well-motivated.
* The approach proposed in this paper is sensible and seems sound: dynamically maintain the coreset from the static algorithm of Jourdan et al.
* The theoretical guarantees seem favorable in terms of the tradeoff between update time, query time, and size of coreset.
* The algorithm has been implemented and compared against a few baseline methods. The new algorithm does exhibit runtime advantages in the experiments considered.

Overall, I like a lot of aspects of this paper and would be happy to see more papers of this flavor at ICLR.

**Weaknesses:**

* The paper relies heavily on the results of Jourdan et al. Building on prior work is not in itself an issue, and the fact that this paper focuses on dynamic algorithms while Jourdan et al just focused on the static case shows that there is clearly something new here. At the same time, the task of making the sampling trees work in a dynamic setting is a somewhat modest contribution. The main technical contribution seems to take up just 3 pages in the main text (the just in-time sampling trees), and a large percentage of the paper is just recounting prior work.

* The experimental results are not that extensive, and only consider a few datasets (only two of them real world datasets).

* There are several things that are pretty unclear regarding the theoretical results, relating to how well k-means and normalized cut are being solved and exactly what approach to spectral clustering is being used. The informal theorems talk about algorithms for normalized cut as if this could be computed or approximated efficiently, but there isn't even a constant factor approximation known for this problem. The best known approximation factor is \sqrt{log n} for ratio cut problems like this (due to Arora Rao and Vazirani). So when Theorem 2 (informal) states that "There exists a dynamic data structure for normalised cut", I'm not quite sure what this could mean because in some sense there isn't an appropriate data structure and algorithm for normalised cut even on static graphs. Similarly, when describing the work of Jourdan et al, this paper says, "Following this, again via the equivalence, they solve the corresponding normalised cut
problem on the coreset graph to get the coreset partition". Are they optimally solving normalized cut? My guess is that they are not, but are instead only approximating it somehow. Can you give more precise details what is happening here?

Furthermore, there is more than one approach people have used for spectral clustering---is the idea to create the spectral embedding and then run k-means, or are you doing some type of a sweep cut? If k-means, then what algorithm for k-means are you using, and since the NP-hard k-means objective is likely not being solve to optimality, what impact does this have on your theory results? Also regarding spectral clustering: are you using the normalized Laplacian or the standard Laplacian? Either way, the theoretical approximation ratio achieved by spectral clustering tends to not be that good in the worst case. All in all, I'm not quite sure how to understand Theorems 1 and 2 in the introduction in light of all these questions and previous results. I'm hoping there is a clear answer to all of these questions, but it's not clear right now.

Looking in Appendix A helps answer some of these questions (e.g., we see in Theorem 3 that we are actually talking about approximations for normalized cut, and not exact solvers), but doesn't answer all questions. Line 5 in Algorithm 4 just saying "spectral clustering", but this does not answer the questions listed above. Furthermore, I'd encourage the authors to distinguish more clearly even in the main text the difference between solving normalized cut and approximating normalized cut.

As a final minor note, in the appendix there is a section header (B Complexity Comparison) that is empty.

**Questions:**

First of all, see questions in the weaknesses section regarding how well normalized cut and  k-means are solved, and what exactly is being done for the spectral clustering.

Another question is about Figure 1: in the last two columns of plots have a symmetric behavior where runtimes start good and get worse and then go back, and ARI starts good and then goes down and then goes back again. Can you explain a little bit more what is going on here? Why the nearly symmetric behavior?

---

> ### Author Response · Authors · 2025-11-19
>
> Thank you for your questions and comments.
>
> **the task of making the sampling trees work in a dynamic setting is a somewhat modest contribution**
>
> Sampling trees themselves are already dynamic. The significant contribution of this paper is the decomposition of $f$ and $g$ over the nodes of the sampling tree so that the tree can be updated and sampled from efficiently.
>
> **The experimental results are not that extensive, and only consider a few datasets**
>
> We extended the experimental setup of Laenen & Sun (2024), and used the same datasets.
> Of course, further experimental evaluation is always desirable and we are open to suggestions.
>
> **What is meant by Normalised Cut?**
>
> To avoid restating Jourdan et al. more than necessary, we deferred the discussion about approximation guarantee to Appendix A. You are right to wonder how we or even Jourdan et al. "solve" the normalised cut problem.
>
> Firstly,
> the version of normalised cut we use is not the standard normalised cut as it has been shifted by additive and multiplicative constants ($\text{tr}(D^{-1}A)$ and $k$ respectively). This comes from the correspondence between kernel $k$-means and normalised cut (Dhillon et al. 2004, Jourdan et al. 2025).
> Secondly, we (and Jourdan et al.) use any $\alpha$-approximation algorithm for normalised cut as a black box, and this doesn't have to be spectral clustering.
>
> We will include more of this discussion in the body of the paper in the next version.
>
> **since the NP-hard k-means objective is likely not being solve to optimality, what impact does this have on your theory results?**
>
> There are two cases to consider.
> - for constant k, one can solve normalised cut on the coreset graph with $\alpha = 1$ in constant time.
> - otherwise, $\alpha$ is whatever approximation spectral clustering gives you, and it is common to use a constant-factor approximation algorithm for $k$-means.
>
> In either case, our algorithm achieves a $O(\frac{1+\epsilon}{1-\epsilon}\alpha)$-approximation for the normalised cut problem on the original graph.
>
> **B Complexity Comparison)**
>
> Thanks. Table 2 has floated away from this section. We will fix this in the next version.
>
> **Figure 1 Symmetry**
>
> In the MNIST/ EMNIST runs, we first add ground truth clusters (in a random order) from each dataset (in smaller batches of the edges incident to their respective nodes). Then we reverse this process by removing edges in a first in, last out fashion. This is why the ARI is symmetric.

---

> ### Comment · Reviewer_WXXY · 2025-11-20
> **Thanks**
>
> Thanks for the clarifications and details! The answers help clarify my points; I think clarifying what is going on earlier in the manuscript is going to be an important update to make. Overall I'd still like to keep my current score. The fact that experiments follow the setup of Laenen & Sun (2024) is well and good, but there are also lots of graph databases out there with more graphs that could be considered for spectral clustering experiments. The overall contribution would be boosted if the paper went beyond just repeating the experimental setup from an existing paper, especially when there are also concerns about how extensive the main theoretical contribution is over existing work.

---

### Official Review · Reviewer_QK9d · 2025-10-26

**Soundness:** 3
**Presentation:** 3
**Contribution:** 2
**Rating:** 2
**Confidence:** 3

**Summary:**

This paper studies the problem of designing graph clustering algorithms in the dynamic setting. Specifically, such algorithms typically maintain a data structure that is updated upon edge/node insertions/deletions, allowing it to support cluster membership queries at any time. The goal is to achieve small update and query times while maintaining a good clustering quality. Prior dynamic approaches either support insertions only (e.g., Laenen & Sun, 2024) or lack explicit approximation guarantees for the returned clusters.

The main contribution of this paper is a fully dynamic coreset based spectral clustering algorithm that supports both edge and node insertions and deletions. The proposed algorithm achieves an update time of $O(\log n)$ and a query time of $O(d_\textup{max}^2\log ^3(n)+\textup{vol}(Y))$ when querying the cluster memberships of all nodes in a set $Y$. The approximation guarantee for the returned clusters is somewhat unclear, as the authors only provide an informal statement of the main result without specifying the approximation ratio.

Technically, the algorithm builds upon the static coreset spectral clustering algorithm of Jourdan et al. (2025), adapting it to the fully dynamic setting by introducing a just-in-time sampling tree that efficiently supports updates. The paper also includes experimental results comparing the proposed method against multiple baselines, demonstrating improvements in both update and query efficiency.

**Strengths:**

* The paper studies the problem of designing dynamic graph clustering algorithms, which is an interesting and relevant research direction.

* It proposes a fully dynamic coreset spectral clustering algorithm that supports both edge and node insertions and deletions. For a query set $Y$ of size $o(n)$, where $n$ is the number of vertices of the graph, the algorithm achieves a sub-linear query time. This paper claims that the method enjoys good theoretical guarantees on clustering quality, although the explicit approximation ratio is not stated.

* The paper conducts extensive experiments comparing the proposed method with several existing approaches, which is a good aspect of the work.

**Weaknesses:**

* The main theorem is presented only informally, with no explicit statement of the approximation ratio, and it remains unclear how the update and query times depend on this approximation parameter, which weakens the theoretical rigor.

* The contribution of the paper appears somewhat incremental, as it is closely related to the static coreset spectral clustering algorithm proposed by Jourdan et al. (2025), with only limited modifications. Moreover, the paper references the generic framework by Henzinger & Kale (2020) for transforming static coresets into dynamic ones, suggesting that the proposed approach mainly combines ideas from these two prior works.

**Questions:**

* As mentioned before, the paper provides only the informal version of the main result (Theorem 2), while the formal version seems to be absent. What is the theoretical guarantee of clustering quality? If an approximation ratio exists, what is the trade-off between the approximation ratio, update time, and query time?

* Could you clarify what the "Adjusted Rank Index" shown in Figure 1 represents?

* On lines 71～73, this paper mentions "…while maintaining a good quality of clustering (see Section 5 for details of how this is measured)". Could you clarify how the clustering quality is measured, as I could not find this in Section 5?

* In line 4 of Algorithm 2, the denominator is $\varepsilon^4$, whereas in Jourdan et al. (2025) it is $\varepsilon^2$ (line 9 of Algorithm 5 in Jourdan et al. (2025). Is this a typo, or is there a particular reason for this modification?

* The paper appears to focus mainly on minimizing the normalized cut objective. Typically, works on graph clustering assume that the input graph has a good $k$-cluster structure. If the graph lacks such structure, the cluster assignment of a queried node may not be meaningful. Have the authors considered a setting similar to Laenen & Sun (2024), where the input graph is assumed to possess a clear cluster structure and the goal is to design a fully dynamic clustering algorithm under this assumption?

**Typos:**

* lines 62, 132, 142, 361 and many other places: $d_{avg} \rightarrow d_{\textup{avg}}$, $d_{max} \rightarrow d_{\textup{max}}$

* line 160: k-means $\rightarrow k$-means

* line 193: ... and matrix $D$ let $K=\dots \rightarrow$ .. and matrix $D$. Let $K=\dots $

* line 352: ... neighbours where $f(x,y)<f(x,C)\rightarrow f(x,\\{y\\})<f(x,C)$

* line 400: missing a period after equation 12

**Suggestions:**

* the capitalization in parentheses after "Definition" is inconsistent (the one in Definition 1 starts with "kernel" in lower case, while the one in Definition 3 starts with "Graph" in upper case) and should be standardized.

---

> ### Author Response · Authors · 2025-11-19
>
> Thank you for your comments and feedback.
>
> **What is the theoretical guarantee of clustering quality? If an approximation ratio exists, what is the trade-off between the approximation ratio, update time, and query time?**
>
> The approximation factor is $O(\frac{1+\epsilon}{1-\epsilon} \alpha)$, where $\alpha$ is the approximation factor for a normalised cut algorithm, defined in Appendix A. The update time of our algorithm is independent of $\epsilon$, and the query time incurs a factor of $\tilde O(\epsilon^{-4})$.
>
> We present our main theorem informally because the approximation guarantee matches that of the static algorithm of Jourdan et al. (2025), Theorem 3 in Appendix A, exactly. As such, we focus on the part that is different in the body of the paper, the superior update and query running times, and defer the definitions required to understand the formal approximation guarantee to Appendix A, which are identical to those found in Jourdan et al. (2025).
> We will clarify this and add a complete formal statement of Theorem 2 to the appendix.
>
> **The contribution of the paper appears somewhat incremental**
>
> We will add a clearer discussion of the difference between the approach of Henzinger & Kale and ours. In short, Henzinger & Kale maintain a tree where every node in the tree represents a coreset of its children. This inevitably introduces a factor of $\log n$ into the coreset size, with consequences for both update and query time. Our work can be treated as a new framework for maintaining dynamic coresets, and is not based on the technique of Henzinger & Kale.
>
> **Could you clarify what the "Adjusted Rank Index" shown in Figure 1 represents?**
>
> ARI, Adjusted Rand Index, Rand (1971) (mentioned line 435), is a metric for measuring clustering performance, without having to compute an optimal mapping between the predicted partition and a ground truth partition.
>
> **Could you clarify how the clustering quality is measured, as I could not find this in Section 5?**
>
> Following Theorem 3, cluster quality refers to the objective of the normalised cut objective given in Appendix A. We will add clarification to make this clear. In practise, when ground truth is available, ARI can be used instead.
>
> **line 4 Algorithm 2 typo**
>
> There is a typo in line 9 of Algorithm 5 in Jourdan et al. (2025). From line 4 of Algorithm 3 of Jiang et al. (2024), $\epsilon^2$ should be an $\epsilon^4$, which is consistent with our Algorithm 2.
>
> **Clusterability Assumptions?**
>
> This is an excellent question and worth further exploration.
> One key direction would be to show that our coreset algorithm preserves the eigengap assumptions used by Laenen & Sun (2024) (among other works on spectral clustering).
> Then it may be possible to combine techniques from both regimes.
>
> Thanks for all the typos!

---

> > ### Comment · Reviewer_QK9d · 2025-11-23
> >
> > Thank you for the authors’ responses. The clarifications address my earlier concerns. Based on the additional information provided, I am raising my score to $4$. That said, I still find the paper borderline: in my view, the novelty and overall contribution remain limited, and certain aspects—such as the presentation of the main theorem—would benefit from further refinement to improve clarity and strengthen the work.

---

### Official Review · Reviewer_5QPY · 2025-10-30

**Soundness:** 3
**Presentation:** 2
**Contribution:** 3
**Rating:** 6
**Confidence:** 4

**Summary:**

Given a graph $G$, the goal of spectral clustering is to partition the nodes of $G$ into $k$ clusters $C_1,\dots, C_k$ that minimize the average conductance $1/k\sum_{i=1}^k \Phi(C_i)$. It is known that this problem is the dual of the kernel $k$-means problem. The paper presents a dynamic algorithm for maintaining a spectral clustering of $G$ under edge insertions and deletions, with $O(\log n)$ update time and $O(d_{\max}^2 \log^3 n)$ query time (see Theorem 2, line 140, for a more detailed query time).

A standard technique for computing spectral clustering is to use coresets. Roughly speaking, a coreset is a reweighted subgraph $H$ that approximately preserves the kernel $k$-means objective for every $k$-node subset of $G$ (see Definition 2 on page 4 for a detailed definition). The algorithm iteratively uses coresets as follows. At each iteration, it computes a coreset $C$ and then independently samples $O(k^2 /\varepsilon^4 \log^2 k)$ nodes not belonging to $C$ according to a distribution. The algorithm then recurses on the union of the sampled nodes and the coreset $C$ until a desired size is reached.

The main challenge in the fully dynamic setting is that the sampling distribution may change after updates. The coreset $C$ is constructed starting from a node with maximum degree in $G$, and additional nodes are added according to Algorithm 3. After a sequence of edge insertions or deletions, both the maximum degree and the corresponding node may change, thus necessitating an update to $C$ to correctly maintain it. Moreover, the sampling of nodes outside $C$ exhibits a similar sensitivity to updates.

To maintain these changes, the paper uses sampling trees, whose leaves correspond to graph’s nodes, and each non-leaf vertex accumulates the assigned values to its children. The process of computing $C$ then starts at a leaf corresponding to a node in $G$ with maximum degree and proceeds upward, adding vertices one by one until $C$ is constructed at the root. The main idea of the paper is to use these sampling trees dynamically so that, after each update, $C$ (and related structures) can be rebuilt using a standard bottom-up traversal that starts from a leaf corresponding to a node in $G$ with maximum degree.

Target Audience: Anyone interested in (but not limited to) dynamic graph algorithms, clustering, and spectral sparsification.

Recommendation summary: The paper studies spectral clustering in the dynamic setting, which is a well-motivated problem both theoretically and practically. However, as reflected in the review, the paper could be written more clearly. Some parts (such as the time complexity analysis) are overly complicated, while certain crucial aspects, such as the discussion of recourse in Algorithm 1, deserve more attention.

**Strengths:**

– Many real-world applications are dynamic in nature, thus studying spectral clustering in the dynamic setting is well-motivated.
– The paper provides experiments to support its theoretical results.

**Weaknesses:**

– The paper could be written more clearly; see the detailed technical comments at the end of this review for more details.
– It seems that the notation is overly complicated. In particular, the description of how updates affect the sampling rates is unnecessarily complex. The same applies to the explanation of how sampling trees are updated, even though this process follows a fairly standard leaf-to-root update procedure.
– The proposed algorithm is theoretically faster than the trivial approach (i.e., recomputing from scratch after each update) only when $d_{\max} = o(\sqrt{n})$.
– The experiments are limited to small synthetic networks, with a maximum of 1000 nodes.
– The paper does not reference some recent theoretical results on spectral sparsification beyond graphs. For example, recent work on spectral hypergraph sparsification in the dynamic setting includes https://arxiv.org/abs/2502.03313 and https://arxiv.org/abs/2502.01421.

**Questions:**

On line 376, why is $C(x^\star)= X$? Could it be that, for some $y\in X$, $\Delta(x^\star, x^\star)\neq \Delta(y, x^\star)$, so that $y\notin C(x^\star)$?

**Detailed Technical Comments**

– page 1, abstract: abstract is vague and does not clearly summarize the paper. e.g., what is $Y$?
– page 3, table 1: please specify whether the algorithms are fully dynamic or partially dynamic (i.e., incremental or decremental).
– line 546: is not conductance usually defined based on $\min\{S, V\setminus S\}$?
– line 588: appendix B is missing.
– Please clarify earlier in the paper that $x^\star$ is a node with the highest degree. The definition on line 1 of Algorithm 3 may confuse readers.
– line 309: please emphasize that $C$ in the equations refers to  $C$ before updates.
– lines 310-316: what is the goal of these calculations? Is it not obvious from the definition that we have quotation (3) immediately?
– line 347: for the summation in the middle, please put parentheses around the entire summed term.

---

> ### Author Response · Authors · 2025-11-19
>
> Thank you for your detailed comments and feedback.
>
> **The notation is overly complex**
>
> While updating a sampling tree is conceptually straight-forward, decomposing the values maintained at each node so we can quickly update and sample from the coreset distribution is complicated, hence the complex notation.
>
> **The proposed algorithm is theoretically faster than the trivial approach only when $d_{\max} = o(\sqrt{n})$**
>
> This is True!
> There are a couple of techniques which could improve this. Firstly, one could employ a dynamic cluster preserving sparsifier (Laenen & Sun (2024)), which will bring the degree of the graph down while preserving our objective
> Secondly, for each vertex in the graph, one can maintain the neighbour list sorted ascending by kernel distance. This means we don't have to read the full neighbour list in line 87 of Data structure 1. Combined with suitable assumptions on kernel space, this could be used to break the quadratic dependence on $d_{\max}$.
>
> **The experiments are limited to small synthetic networks, with a maximum of 1000 nodes.**
>
> This is not true.
> The confusion might be that for our real world workloads, we add updates in batches of 1000 nodes. The EMNIST dataset has over 130,000 nodes.
>
> **Question about line 376**
>
> Since there is only one seed present in $C$ after line 2 of Algorithm 3, there can only be a single seed set at that point.
> In other words, for every point in $X$, $x^*$ is its closest seed. Notice that the definition of seed sets uses $\mathrm{arg min}$ - it is the set of points who share a closest seed point.
>
> **Technical Comment about $Y$**
>
> $Y$ is a set of nodes to be labelled. The time to label it depends also on its size. We can make the wording slightly clearer in the abstract. It matches the definition of $Y$ in Table 1.
>
> **Re fully dynamic or not in Table 1**
>
> We will add an extra column saying which setting each algorithm works in.
>
> **Conductance comment**
>
> Yes you are correct. We are using the definition from Dhillon et al., 2004, Section 2.2.1.
>
> **Appendix B**
>
> It appears Table 2 has floated below Appendix B (where it was supposed to be). We will fix this in the next version.
>
> **Other technical comments**
>
> Many thanks for your suggestions. We will make some clarifying edits in the next version.

---

> > ### Comment · Reviewer_5QPY · 2025-11-23
> >
> > Thanks for your detailed response. After reading the other reviews (especially the issue of making the theorem more formal), as well as conducting additional experiments and perhaps removing the assumption on the degree, I would say that the paper requires another serious pass before it's fully ready for publication. Hence, I would like to keep the same score.

---

### Official Review · Reviewer_HQRM · 2025-10-30

**Soundness:** 3
**Presentation:** 2
**Contribution:** 3
**Rating:** 4
**Confidence:** 3

**Summary:**

This paper proposes a dynamic sampling tree that can be used in constructing coresets for kernel k-means and then dynamic coreset spectral clustering. They carefully decompose and analyze the maintenance of the weighted distance of a point to the coreset (f function) and the sum of relative weighted distance and relative weight (g function) undergoing graph edge updates. Applying the technique into the recent coreset spectral clustering yields dynamic coreset spectral clustering with logarithmic update time and sublinear query time. Experiments are performed on comparing against several baselines on update time, query time, and clustering quality.

**Strengths:**

Strengths:
1. The paper studies an interesting problem of dynamic coreset spectral clustering, extending recent static coreset spectral clustering Jourdan et al. (2025). The improvements on update time and query time are obvious and significant.
2. The experimental evaluations on real-world and synthetic workloads are interesting, well supporting the proposed theoretical analysis.
3. The introduction section is well-written, providing a good overview on the background and the proposed technique.

**Weaknesses:**

Weaknesses:
1. My main concern is the presentation of the technical part. It is quite unfriendly to readers that are not familiar with coreset k-means techniques. The authors may want to enhance the writing to help people appreciate the technical contributions. For example, formal proof for the theorems should be provided.
2. The performance comparison in Table 1 ignores the approximation factor due to the nature of coresets. It would be best to explicitly state that the approximation of the dynamic coreset technique in spectral clustering while enjoying the improvements in the update and query time.

Comments:
- The merge & reduce technique in the previous dynamic coreset k-means in Henzinger & Kale (2020) and the relationship with the proposed dynamic coreset algorithm should be made clear.
- Related to the above comment, related work on k-means coreset construction and their dynamization could make the literature review more comprehensive.
- Line 80: Once getting a newly maintained coreset, the algorithm runs the static coreset spectral clustering of Jourdan et al. (2025). The computational complexity of spectral clustering should be provided directly, since it is not clear by only looking at Table 1.
- Tables 1 and 2: when polylog factors are ignored, should use a different notation compared to big-O (e.g., with tilde of O).
- Line 418: An intuition of the definition of D can be provided to enhance the readability.
- The different definitions in Section 2 can be simplified and improved for better readability.
- ARI has a maximum value of 1.0 and the portion above 1.0 in Figure 1 due to standard deviations can be cutoff to avoid misunderstanding.

**Questions:**

Please see the weaknesses above.

---

> ### Author Response · Authors · 2025-11-19
>
> Thank you for your comments and feedback.
>
> **For example, formal proof for the theorems should be provided**
>
> Our main theorem is stated only informally because the approximation guarantee matches that of Jourdan et al. (2025), Theorem 3 in Appendix A, exactly. As such, in the body of the paper we focus on the part that is different: the superior update and query running times.
> We include the formal approximation guarantees in Appendix A, which are identical to those found in Jourdan et al. (2025). We believe that this makes the paper more readable.
>
> **The performance comparison in Table 1 ignores the approximation factor.**
>
> The algorithms listed in Table 1 are not directly comparable in terms of the approximation guarantee. Specifically,
> Laenen & Sun (2024) target a different objective to ours, and moreover the naive algorithm is a heuristic without a formal approximation guarantee.
> The other methods all achieve the same approximation guarantee for the normalised cut objective. We will clarify this in the next version.
>
> **The relationship with the proposed dynamic coreset algorithm [and Henzinger & Kale (2020)] should be made clear.**
>
> We will add a clearer discussion of the difference between the approach of Henzinger & Kale and ours. In short, they have to maintain a tree where every node in the tree is a coreset of their children. This inevitably introduces a factor of $\log n$ into the required coreset size, with consequences for both update and query time.
>
> **Related work on $k$-means coreset construction**
>
> We will expand our discussion of related work on k-means coreset construction and the dynamization. One reason why such techniques are hard to apply in our setting is that the underlying metric space changes over time.
>
> **The computational complexity of spectral clustering should be provided directly**
>
> The computational cost of running spectral clustering on the coreset graph is dominated by sampling the coreset and so it is included in the complexities given in Table 1.
>
> **The different definitions in Section 2 can be simplified and improved for better readability.**
>
> We are open to simplifying and clarifying our exposition, and are interested if there is a specific definition which you feel could be improved.
>
> Many thanks for your other minor comments and suggestions. We will make some corresponding edits in the next version.

---

> > ### Comment · Reviewer_HQRM · 2025-11-22
> >
> > Thank you for the response. It appears that multiple reviewers had the same concerns on the rigor of the informal main Theorem and the presentation, such as the approximation factor. It would also be great if a formal proof can be provided explicitly. Considering the technical novelty of the theory and the presentation of the current manuscript, I'd like to keep my score.

---

### Meta-Review · Area_Chair_5Qkc · 2025-12-28

**Summary:**

The reviews are negative overall. The concerns raised by the reviewers are valid. The authors' response is only able to partially resolve the issues. I suggest to reject the paper after all.

**Reviewer Concerns:**

The following are the outstanding issues, albeit they may be partially addressed in the rebuttal.

1. The experiments can still be improved, by adding more datasets etc.
2. The clarity of the technical proofs.
3. The new method does not seem to be new, and hence the novelty is limited.

Other issues are resolved, and are acknowledged by the reviewers.

**Reviewer Scores:**

The reviewers already expressed their preference of keeping the scores.

---

### Decision · Program_Chairs · 2026-01-26

Reject